# Versatile Double Bandgap Photonic Crystals of High Color Saturation

**DOI:** 10.3390/nano13192632

**Published:** 2023-09-24

**Authors:** Hao Jiang, Gang Li, Luying Si, Minghui Guo, Huiru Ma, Wei Luo, Jianguo Guan

**Affiliations:** 1State Key Laboratory of Advanced Technology for Materials Synthesis and Processing, International School of Materials Science and Engineering, Wuhan University of Technology, Wuhan 430070, China; jh1998@whut.edu.cn (H.J.); siluying@whut.edu.cn (L.S.); guanjg@whut.edu.cn (J.G.); 2School of Materials Science and Engineering, Wuhan University of Technology, Wuhan 430070, China; chemligang2010@whut.edu.cn (G.L.); gmh1232000@163.com (M.G.); 3Wuhan Institute of Photochemistry and Technology, 7 North Bingang Road, Wuhan 430083, China; 4School of Chemistry, Chemical Engineering and Life Science, Wuhan University of Technology, Wuhan 430070, China

**Keywords:** double bandgap, color modulation, color saturation, photonic nanochains, pigments

## Abstract

Double bandgap photonic crystals (PCs) exhibit significant potential for applications in various color display-related fields. However, they show low color saturation and inadequate color modulation capabilities. This study presents a viable approach to the fabrication of double bandgap photonic inks diffracting typical secondary colors and other composite colors by simply mixing two photonic nanochains (PNCs) of different primary colors as pigments in an appropriate percentage following the conventional RGB color matching method. In this approach, the PNCs are magnetically responsive and display three primary colors that can be synthesized by combining hydrogen bond-guided and magnetic field (*H*)-assisted template polymerization. The as-prepared double bandgap photonic inks present high color saturation due to the fixed and narrow full-width at half-maxima of the parent PNCs with a suitable chain length. Furthermore, they can be used to easily produce a flexible double bandgap PC film by embedding the PNCs into a gel, such as polyacrylamide, facilitating fast steady display performance without the requirement of an external magnetic field. This research not only presents the unique advantages of PNCs in constructing multi-bandgap PCs but also establishes the feasibility of utilizing PNCs in practical applications within the fields of anti-counterfeiting and flexible wearable devices.

## 1. Introduction

Numerous organisms exhibit striking and vibrant hues within their natural habitats, with certain colors stemming from an amalgamation of diverse pigments [1,2,3], while others arise from structural coloration, as observed in cuttlefish [4], chameleons [5], beetles [6] and so on [7,8]. The material’s microstructure is responsible for its structural coloration, interacting with incident light, thereby impeding the transmission of specific frequencies and resulting in pronounced reflection. Photonic crystal (PC) structures, exemplifying a quintessential manifestation of structural coloration, exhibit a photonic bandgap (PBG) stemming from the periodic arrangement of two or more dielectric materials. Within this PBG, electromagnetic waves bearing energy are unequivocally precluded from propagating through the PC matrix. Consequently, structural coloration possesses distinct properties in comparison to chemical coloration, notably including angle dependence [9,10,11], non-photobleaching [12,13,14], and a wide array of research and application prospects encompassing the fields of display technology [15,16,17,18,19], anti-counterfeiting [20,21,22,23], sensing [24,25,26,27,28] and bionics [29,30,31,32]. 

The colors perceptible in our everyday surroundings typically arise from the combination of multiple light frequencies with varying intensities. It is evident that monochromatic PCs lack the inherent capacity to engender composite colors unless disparate instances of single-bandgap PCs are conjoined. In recent years, the realm of multi-bandgap PCs has elicited notable scholarly consideration, with a typical way of stacking the distinct single-stopband 3D PCs to fabricate materials endowed with multiple bandgaps [33,34,35]. For example, Kim and coworkers first prepared single photonic bandgap films with three primary colors by using differently sized silica colloidal particles (SiO_2_).Based on these, they constructed double bandgap PC films by stacking two of three primary colors PC films in an equal thickness. Both the single and double-stopband PC films displayed vibrant single and secondary structural colors correspondingly, due to their comprehensive consideration of the similar refractive index between SiO_2_ and resin matrix to remove the disadvantageous impact from stray light and interfacial reflection. Wang et al. [36] prepared multi-stopband PCs by compositing the biopolymer matrix with 3D inverse opal structure formed by layer-by-layer stacking and subsequently removing colloidal crystal monolayers based on polystyrene (PS) spheres with two different sizes. The bandwidth and corresponding structural color of multi-stopband PCs can be controlled by tuning the arrangement patterns of two monolayers within them. Though layer-by-layer stacking approaches have been demonstrated feasible to construct multi-stopband PCs, the time-consuming assembly and stacking process, and the required exact match of optical properties between the matrix and building blocks will make them face a great challenge when applied in some fields. A facile and expeditious approach to constructing multi-stopband PCs for obtaining on-demand structural color should involve the direct mixing of colored units as that of the generally used tinting method. The Takeoka group [37,38] reported photonic pigments characterized as sub-micron-sized monodispersed photonic balls composed of silica particles, which can diffract three primary colors when 221, 249, and 291 nm silica nanoparticles are employed. Various colors were reproduced by tuning the mixture ratio of photonic pigments. Besides, a black background and a specific size of photonic ball are required for producing a high color saturation and bright structural colors. Diiferent from polymer colloidal crystals, magnetically responsive photonic crystals (MRPCs) exhibit bright structural color due to their high refractive index and rapid response to external magnetic field. Both the Hu [39] and Shang [40] groups reported multi-bandgap PCs by blending single-bandgap MRPC according to the three primary colors of light and possible forming principles of multi-bandgap were also investigated. However, optical property results show that the spectra of certain mixtures deviated from their constituent spectra, thus challenging adherence to the additive color law. Mixtures demonstrating a dual bandgap often exhibited reduced color saturation in the corresponding bandgap compared to their pre-mixing state. This phenomenon may be attributed to the fact that superparamagnetic nanoparticles from initially different single-bandgap MRPC inevitably interfere with each other during the magnetic assembly process, which brings an irregular arrangement of nanoparticles in one-dimensional chainlike structures. As a result, the current construction strategy of multi-bandgap PC materials and corresponding gain of on-demand colors still remain unsatisfactory.

In this study, we propose a viable methodology for the fabrication of double bandgap PCs utilizing magnetically responsive PNCs as photonic pigments. The PNCs, prepared through a hydrogen bond-guided template polymerization method, exhibited high color saturation and diffracted distinct structural colors. The microstructure and composition of PNCs were characterized using scanning electron microscopy (SEM), optical microscopy, and Fourier transformed infrared (FT-IR) spectroscopy. Double bandgap liquid PCs were successfully created by blending PNCs with different primary colors, and their satisfying photonic property and mixed colors are presented, benefitting from the excellent optical properties and magnetically untunable interparticle distance of PNCs. In addition, the integration of different colored PNC pigments with chemically cross-linked gel was found to be a desirable for fabricating double bandgap flexible PC film. The results reported in this study show that the as-obtained double bandgap PC materials hold significant potential to advance applications in color display, anti-counterfeiting, flexible wearable devices, and other related fields.

## 2. Materials and Methods

### 2.1. Materials

The following materials were purchased from Aladdin Reagent Co., Ltd. (Shanghai, China): 2-hydroxethyl methacrylate (HEMA), ethylene glycol dimethacrylate (EGDMA), 2-hydroxy-2-methylpropiophenone (HMPP), acrylamide (AAm) and ammonium persulphate (APS). Ethanol, dimethyl sulfoxide (DMSO) and ethylene glycol (EG) were purchased from Sinopharm Chemical Reagent Co., Ltd. (Shanghai, China). All chemicals were used directly as received without further purification. Deionized water (18.20 MΩ cm) was produced in a Milli-Q system (Millipore, Burlington, MA, USA). Superparamagnetic Fe_3_O_4_@PVP nanoparticles were prepared by a modified polyol process according to our previous report [41].

### 2.2. Preparation of Fe_3_O_4_@PVP@PHEMA Photonic Nanochains

In a typical synthesis of green (530–540 nm) PNCs, 0.45 mg monodispersed superparamagnetic 150 nm of Fe_3_O_4_@PVP nanoparticles (Appendix A) were mixed with 26.1 mg HEMA, 0.658 mg HMPP, and 4.766 mg EGDMA by ultrasonic treatment to form a homogenous brown solution. Then, 31.974 mg of this homogenous brown solution was dispersed in 0.6 mL EG solvent, and afterward, 0.1 mL deionized water was added to form a prepolymer solution. After that, the prepolymer solution was placed above a rectangular NdFeB permanent magnet (10 × 10 × 2 cm) providing a magnetic field (*H*) strength of 200 Gauss. After 1 min, the UV light source was turned on for 5 min. The product was washed with a 50% ethanol aqueous solution and redispersed in a 50% ethanol aqueous solution. Changing the diameters of Fe_3_O_4_@PVP nanoparticles resulted in Fe_3_O_4_@PVP@PHEMA PNCs diffracting different colors. For example, the red and blue primary color PNCs were obtained by using nanoparticles with a diameter of about 180 and 120 nm, respectively. 

### 2.3. Preparation of Double Bandgap Photonic Crystal Film

The double bandgap PC films were fabricated on a glass substrate through external magnetic field-assisted thermal-initiation polymerization. A 1 mm-thick square gasket cavity (2 cm × 2 cm), used as a mold, was stuck on the glass substrate. In the preparation process of double bandgap PC film, 0.3554 g AAm, 0.015 g EGDMA, and 0.0228 g APS was dissolved in 570 μL PNCs DMSO dispersion to form a prepolymer solution and stored at 15 °C. The PNCs DMSO dispersion contains two kinds of PNCs with different photonic bandgaps, the total weight of PNCs is 1 mg. Afterward, 400 μL of the above precursor solution was quickly injected into the square gasket cavity, and a piece of cover glass was placed on it to form a closed chamber, under which a NdFeB permanent magnet was placed to provide an external magnetic field (*H* = 100 Gs) to guide PNCs orientation vertically. After being heated for 10 minutes at 70 °C in an oven, the double bandgap photonic crystal film was prepared, along with the two primary colors Fe_3_O_4_@PVP@PHEMA PNCs embedded in the PAAm gel.

### 2.4. Characterizations

All digital photos in this paper were taken by iPhone 12 with a dual 12 MP camera system, and the illumination conditions remain 1000 lux. The field emission SEM (FE-SEM) images were captured on a Hitachi S-4800 scanning electron microscope. The FT-IR spectrum was obtained using a Nicolet 60-SXB FTIR spectrometer in the range of 400–4000 cm^−1^ with a resolution of 4 cm^−1^. The thermal analysis of photonic nanochains was carried out with a NETZSCH-STA449C/G instrument under air in the temperature range from room temperature to 1000 °C with a heating rate of 5 °C·min^−1^. The hysteresis loops were acquired by a vibrating sample magnetometer (VSM, Model 4 HF, ADE, Chicago, IL, USA). The reflective spectra were obtained using a fiber optic spectrometer USB 2000+. Optical microscopy images were recorded through an optical microscope (Zeiss Axio Observer 5M, Göttingen, Germany). The initial length of the bar-type sample used for stretching and the stretched length are recorded as the length between the two tweezers. The chain length distribution maps of the PNCs were obtained by measuring the length of 100 PNCs

## 3. Results

Photonic nanochains (PNCs) are currently the smallest colored unit of one-dimensional (1D) PCs, possessing a fixed photonic stopband and bright structural color [42]. Figure 1 illustrates the preparation of Fe_3_O_4_@PVP@PHEMA PNCs by combining hydrogen bond-guided and *H*-assisted template polymerization. Our experimental procedure involved the addition of monomer (HEMA), crosslinker (EGDMA), photoinitiator (HMPP), monodispersed superparamagnetic Fe_3_O_4_@PVP nanoparticles, and deionized water to EG, resulting in the formation of a prepolymer dispersion liquid.

In the precursor liquid, since PVP is rich in a large number of carbonyl groups, the monomer will form a hydrogen bond with PVP as the hydrogen bond donor and will be enriched around the PVP shell of the nanoparticle. The localized concentration of monomers near the nanoparticles exceeded that of other regions within the prepolymer solution. Subsequently, the precursor solution was subjected to magnetic field orientation. Due to the existence of the PVP shell, the nanoparticles were in a state of balance between steric resistance and magnetic force, forming a stable spacing 1D chain template. With the irradiation of UV light, the monomers enriched around the nanoparticles undergo polymerization reactions with crosslinkers and photoinitiators to form a polymer gel covering the chain and finally yielding 1D magnetic PNCs resembling the peapod structure. The monomer (HEMA) used in the preparation process is hydrophilic, which enables PNCs to be dispersed in ethanol aqueous solution and polar solvents such as dimethyl sulfoxide. By changing the size of superparamagnetic particles and the strength of the magnetic field, we can easily prepare the PNCs with the three primary colors.

Figure 2a presents the digital photo and corresponding reflection spectrum of the typical Fe_3_O_4_@PVP@PHEMA PNCs liquid. Thanks to the polarity of the PHEMA [43] shell, these PNCs are well dispersed in a 50% ethanol aqueous solution to form homogeneous dispersion, exhibiting a vibrant green structural color under *H* simultaneously, as shown in the inset, and corresponding to a steep single peak at 532 nm. The reflection spectrum further gives a full-width at half-maxima (FWHM) of 40 nm, suggesting a high color saturation of PNCs. In Figure 2b, the scanning electron microscopy (SEM) image provides insights into the microscopic morphology of the PNCs, revealing the formation of singular peapod-like PNCs. The uniform magnetic nanoparticles are encapsulated and separated by a thin polymer shell, and the average interparticle distance within the nanochain is approximately 190 nm. According to Bragg’s diffraction law *m**λ* = 2*nd*sin*θ* [19], where *m* signifies the diffraction order, *λ* represents the wavelength of reflectance, *n* and *d* are the effective refractive index and lattice distance, respectively, and *θ* is the glancing angle of the incident light. Given the low PNCs concentration, it is reasonable to approximate *n* as the refractive index of the dispersion medium (*n*_dispersion_ ≈ 1.358) [44]. For normal incidence (*θ* = 90°), substituting these values into the aforementioned formula yields an estimated diffracted light wavelength of approximately 515 nm. It is noteworthy that this estimated value is smaller than the reflectance wavelength observed in Figure 2a. This discrepancy may be attributed to the SEM sample preparation process, which necessitates a drying step, and the polymer shell of the chain undergoes some degree of shrinkage, thereby implying that the actual inter-particle spacing within the chain may be marginally larger compared to what is visually observed in the SEM image [41,45]. As previously reported [42], the as-prepared PNCs also show excellent magnetic susceptivity and generate rotational orientation along the magnetic field direction, as shown in Figure 2c. Upon being applied to a parallel magnetic field, almost all PNCs arrange in parallel along the external magnetic field and present a straight configuration. The average length of the PNCs is about 9.2 μm (Appendix A), which eliminates the defects produced by the assembly of short PNCs and thus benefits the narrow FWHM and high color saturation [46]. When a vertical magnetic field was applied, the well-ordered PNCs exhibited bright green dots in dark-field mode microscopy. More importantly, different from superparamagnetic nanoparticle-based PCs that exhibit magnetically tunable structural colors, PNCs display an unchanged diffraction wavelength and structural color with changed magnetic field strength (Appendix A), manifesting the non-reassembly of superparamagnetic nanoparticles within PNCs under a magnetic field and good operability of optical properties. Meanwhile, the repeatedly actuated experiment under a magnetic field (Appendix A) indicates excellent optical stability of PNCs, so it is reasonable for us to believe that PNCs are the ideal optical pigments.

Besides, the chemical composition of Fe_3_O_4_@PVP@PHEMA PNCs was further characterized using the Fourier−transformed infrared (FTIR) spectrum, as depicted in Figure 2d. The FTIR spectrum confirms the presence of characteristic absorption bands, including the carbonyl group (C=O) in PHEMA at 1722 cm^−1^ and C−O−C stretching in PHEMA at 1156 cm^−1^. Peaks at 1651, 1445, and 1268 cm^−1^ correspond to characteristic peaks of PVP. The peak observed at 561 cm^−1^ arises from Fe-O in Fe_3_O_4_. The thermo-gravimetry-differential scanning calorimetry curves (TG-DSC) in Figure 2e exhibit a significant decomposition of organic components during the calcination process up to approximately 400 °C. The weight loss of approximately 33.71% in this stage is more than twice that of the original Fe_3_O_4_@PVP nanoparticles, providing evidence for the presence of a PHEMA shell in the resulting PNCs. In Figure 2f, the saturation magnetization of the Fe_3_O_4_@PVP@HEMA PNCs is determined to be 38.2 emu/g, with negligible coercivity and remanence, as indicated in the inset. These findings collectively confirm the successful fabrication of one-dimensional photonic nanochains comprising periodically arrayed magnetic Fe_3_O_4_@PVP nanoparticles encapsulated by PHEMA polymer coating [42].

The diffraction wavelength and corresponding structural color generated by PCs originate from Bragg diffraction. It means that PNCs with different structural colors can be conveniently obtained by tuning the lattice parameter of magnetic-induced chain-like PC template during the preparation process of PNCs (Figure 1). Based on this, we designed and fabricated the other two primary color PNCs by simply using differently sized Fe_3_O_4_@PVP nanoparticles while keeping other preparation conditions constant. Specifically, compared to the typical green PNCs, monodispersed Fe_3_O_4_@PVP nanoparticles with a larger size about of 180 nm (Appendix A) were utilized to fabricate red PNCs and monodispersed Fe_3_O_4_@PVP nanoparticles with a smaller size about of 120 nm (Appendix A) were used to fabricate blue PNCs. As expected, the resulting PNC-based dispersions in the magnetic field present their diffraction wavelength at 611 nm and 470 nm, respectively, and show bright red and blue accordingly (Figure 3a,d). 

Besides, the relatively narrow FWHM of reflection spectra in Figure 3a,d also reflect their high color saturation [23]. SEM images in Figure 3b,e further demonstrate that diffraction peaks and structural colors in Figure 3a and d are derived from the interaction of PNCs with visible light but not Fe_3_O_4_@PVP nanoparticles-assembled photonic structures. Figure 3c,f give the optical microscopical photos of parallel-orientated PNCs, from which no single nanoparticles are observed. The average chain lengths for these two primary color PNCs are 13 μm and 16.6 μm, as shown in Appendix A. A great many vivid red and blue monochromic pixels are observed in dark field mode under a vertical magnetic field, as shown in the insets of Figure 3c,f. In fact, individual pixels correspond inherently to oriented PNCs. Remarkably, the dimensions of these pixels align precisely with the diameters of the PNCs, thereby conferring upon them an unparalleled advantage in terms of color resolution [42]. This distinctive attribute stands in stark contrast to pigments founded upon three-dimensional photonic crystals with micrometer-scale dimensions.

The CIE-1931 chromaticity diagram (Figure 4a) gives the positions of colors diffracted by the three primary color PNCs prepared above, which intuitively manifests the high color saturation of PNCs and reflects their potential reproduction ability to various colors. Significantly, the unique features involving the same composition, fixed interparticle space, and tiny size as color pixel points make it feasible for these different primary color PNCs to construct multiple-bandgap PC and diffract uniform composite colors through the simple mixing approach in accordance with additive color mixing method. To verify it, the typical double bandgap PC liquids diffracting secondary colors were first built by combining two of three primary color PNCs in equal proportion, and the produced optical properties are shown in Figure 4b–d. Specifically, the combination of red and green PNCs yields yellow (Figure 4b), while mixing red and blue, as well as green and blue, leads to magenta (Figure 4c) and cyan (Figure 4d), respectively. All of the reflection spectra in Figure 4b–d consist of two separated peaks, demonstrating their essential double bandgap characteristics [47,48,49]. And the identical peak height agrees with the 1:1 weight ratio of the two PNCs in the mixture. It also can be observed that each peak in the reflection spectra almost keeps the same shape and peak position as that of their corresponding parent PNCs, suggesting that two primary color PNCs have no negative effects on each other on diffraction to visible light in the mixing liquid (Appendix A). Besides, double bandgap PC liquids diffracting other composite colors were proved to be achievable through adjusting the mixing ratio of red and green PNCs. When blended in more green PNCs, the color diffracted by the constructed double bandgap PC liquid looks greenish and the diffraction peak corresponding to green PNCs dominates in the double peaks (Appendix A). On the contrary, when more red PNCs were added to the mixture, the double bandgap PC liquid diffracted reddish structural color and the diffraction peak from red PNCs became stronger accordingly (Appendix A). Simply put, PNCs can be used as photonic pigments to realize the preparation of double or multi-bandgap PC liquid and can especially conveniently and freely adjust color just like general chemical pigments. 

In practice, PC film is another important application form and has shown bright prospects in the fields such as photonic paper, color sensors, and wearable display devices. The reported multi-bandgap PC films were generally fabricated via a multi-step stacking method and needed support from a substrate. Here, using red and green PNCs as photonic pigments, a free-standing double bandgap PC film was prepared by a one-step process via *H*-assistant free-radical polymerization. Its optical property and flexibility were investigated. Both the consistent yellow, which is the secondary color of red and green, and the appearance of double peaks in the reflection spectrum shown in Figure 5a demonstrate the double bandgap feature of PC film. However, in contrast to Figure 4b, the color in Figure 5a is not bright enough, and the double peaks present a merging trend, all of these suggest that the optical properties of the double bandgap PC film are obviously inferior to that of the corresponding double bandgap PC liquid. This result is ascribed to the fact that internal stress produced during film solidification causes the deformation of PNCs and thus has negative effects on optical properties [28]. Consequently, stiffer PNCs utilized will be in favor of the optical properties of the film. Another merit of the double bandgap PC film is its mechanical property. As depicted in Figure 5b, it exhibits excellent deformability, allowing for arbitrary twisting and stretching. It can even be stretched to 2.6 times its original length before rupture (Appendix A). The commendable flexibility endows the double bandgap PC film with better practicability.

## 4. Conclusions

In conclusion, we successfully synthesized individually magnetic PNCs with sufficient chain length and high color saturation, combining hydrogen bond-induced and *H*-assistant template polymerization. More importantly, a reliable approach was proposed in this paper to construct double bandgap PCs using PNCs as photonic pigments through straightforward blending. The color scheme completely follows the conventional additive color mixing. Double bandgap PC liquids diffracted cyan, magenta, yellow hues, and other composite colors were obtained by adjusting the mixing ratio of the two primary color PNCs used. The optical property reveals that PNCs in the mixed liquid have the ability to diffract visible light synergistically and have no interference with each other’s structural periodicity, paving a feasible and convenient way for fabricating multi-bandgap PCs. Meanwhile, flexible free-standing PC film with double bandgap was developed by combining PNCs mixing liquid with elastic polymer. The PNC-based double bandgap PC materials have the characteristics of flexible color adjustment and simple preparation, which endow them with practical applications in fields such as bionics, display technologies, anti-counterfeiting, and flexible wearable devices. 

## Figures and Tables

**Figure 1 nanomaterials-13-02632-f001:**
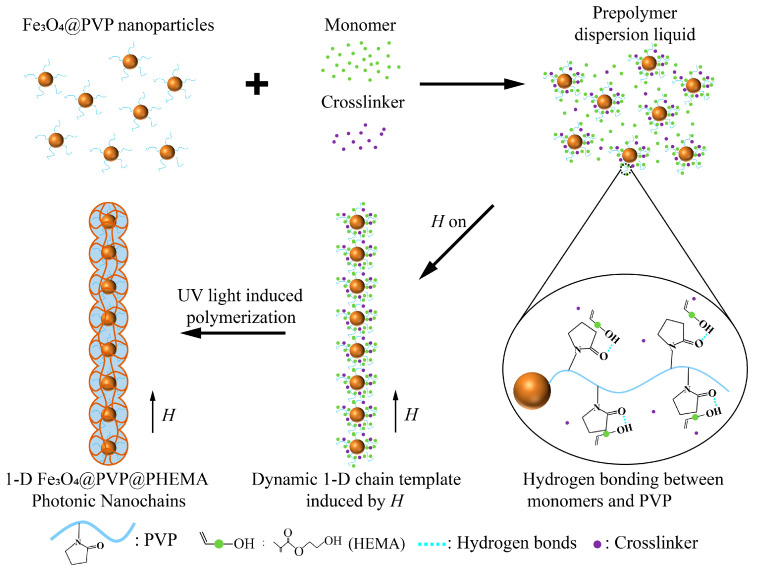
Schematic illustration of the preparation of Fe_3_O_4_@PVP@PHEMA PNCs.

**Figure 2 nanomaterials-13-02632-f002:**
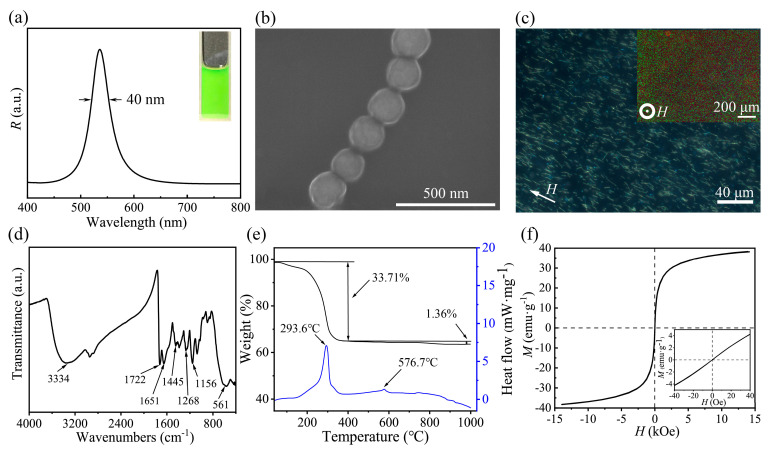
(**a**) Reflection spectrum and digital photo of a dispersion composed of Fe_3_O_4_@PVP@PHEMA PNCs and 50% ethanol aqueous solution under *H* = 200 Gs, (**b**) SEM image of PNCs, (**c**) Dark-field optical microscope images of PNCs under magnetic field, (**d**) FT-IR spectrum, (**e**) TG-DSC curves, and (**f**) Magnetic hysteresis loop of as-obtained PNCs.

**Figure 3 nanomaterials-13-02632-f003:**
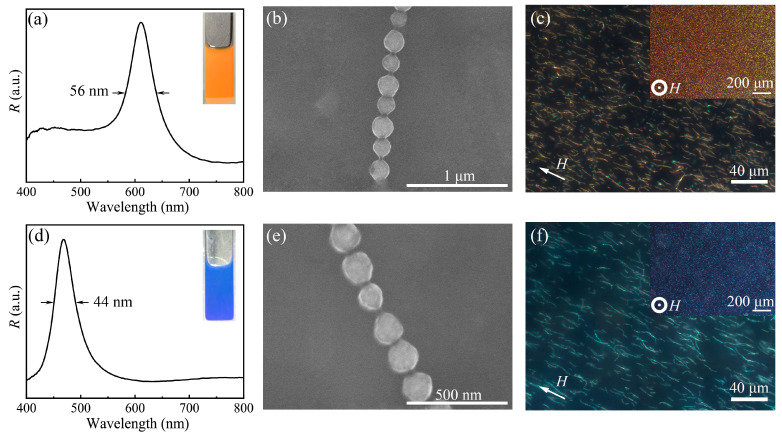
(**a**–**c**) Reflection spectrum, SEM image and dark field optical micro-scope images of red Fe_3_O_4_@PVP@PHEMA PNCs. (**d**–**f**) Reflection spectrum, SEM image and dark field optical microscope images of blue Fe_3_O_4_@PVP@PHEMA PNCs.

**Figure 4 nanomaterials-13-02632-f004:**
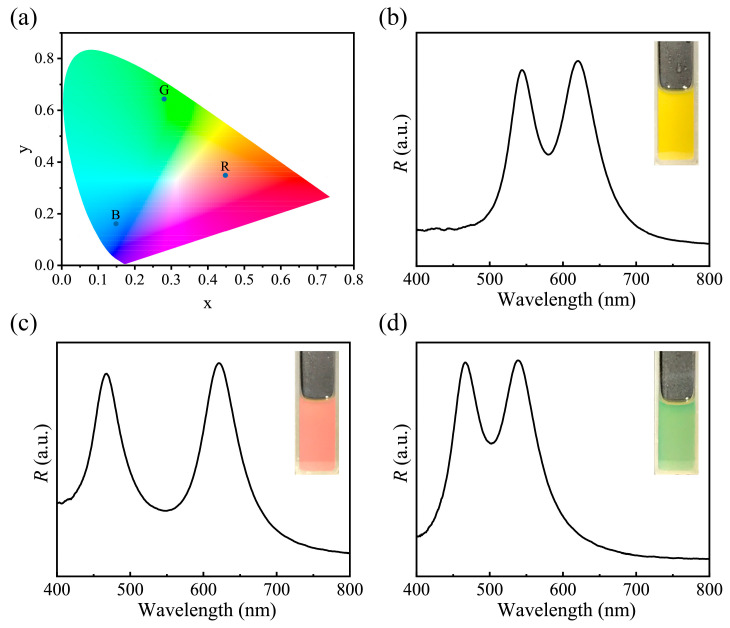
(**a**) CIE-1931 chromaticity diagram of red, green, blue, and their paired blends in equal proportions. (**b**–**d**) The spectra and digital photos of yellow (red PNCs + green PNCs), magenta (red PNCs + blue PNCs), cyan (green PNCs + blue PNCs) photonic liquid.

**Figure 5 nanomaterials-13-02632-f005:**
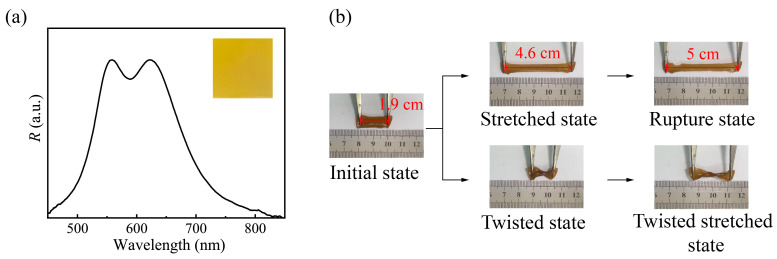
(**a**) The reflection spectrum, digital photo and (**b**) flexibility of PNCs-based double bandgap photonic crystal film.

## Data Availability

The data presented in this study are available on request from the corresponding author.

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
