# Peer review of "Versatile Double Bandgap Photonic Crystals of High Color Saturation"

_nanomaterials, 2023, doi:10.3390/nano13192632_

Round 1

Reviewer 1 Report

*This work addresses the fabrication of inks and filters based on

magnetic polymeric liquids in order to tailor and improve the spectral

light response of these materials and ease their fabrication.

*It represents an improvement of the state of the art of the production

of these materials. As said in my comments in the review more work is

still required to fulfil the market demands of these materials.

*It mainly adds an improvement of the fabrication processes, which

nowadays still rely on critical fabrication parameters also proposes

improvements in the optical response of the obtained materials.

Reviewer 2 Report

There are places in the manuscript where more detail needs to be provided, so the information is useful to the reader, like:

line 22 -  due to the fixed and perfect photonic stop band of the 22
parent PNCs with a long length. - What do they mean by long length? What is a perfect photonic stop band?

line 70 - What is a photonic ball?

line 141 - Which iPhone? What are the specifications of its camera?

line 143 - Include the company name in the FTIR model: "Nicolet 60-SXB FTIR"

 Caption of Fig. 2 - (f) Magnetic hysteresis loop of as-obtained PNCs. - I don't see any hysteresis loop in the graph!

line 181 - "Thanks to the polar from PHEMA shell" - The polar what? From or "of the" PHEMA shell?

Style suggestions:

line 38 - Consider using "properties" in place of "attributes".

line 40 - "fields of display technology, ..." in place of "realms of display, ..."

There are many English mistakes and some phrase misconstructions, that make the author's intended message hard to understand or dubious. A few examples are (non-exhaustive list):

line 24 - embedding the PNCs into a gel of, such as, polyacrylamide, - Either the "of" needs to be cut out or the authors need to specify the "what" it is a gel of...

line 27 - practical applications within anti-counterfeiting field 27
and flexible wearable devices. - Would be better "practical applications within the fields of..."

line 35 - The material responsible for structural coloration employs its microstructure to... - The material doesn't employ its microstructure. It is not an animate being. The material's microstructure is responsible for its structural coloration.

line 45 - "Recent years ..." should be "In recent years ..."

line 53 - "refracture index" should be "refraction index"

line 62 - "optical property" should be "optical properties"

line 94 - should read "excellent optical properties"

line 96 - should read "found to be a desired way"

line 119 - remove the word "last": "was turned on last for 5 min."

line 121 - should read "Changing the diameters of the Fe3O4@PVP nanoparticles resulted in Fe3O4@PVP@PHEMA PNCs diffracting different colors.

line 122 - remove the word "other": "For example, the other red"

line 154 - "This section may be is divided"

line 154 - "subheadings. It should, which provide"

line 190 - "Optical properties"

line 225 - "n and d signify are the effective"

line 226 - " θ signifies is the glancing "

Reviewer 3 Report

It is an exciting paper, and I recommend it for publication once the authors answer the following requests:

1) A deeper explanation of why the photonics effect occurs is needed. In ref 41 from the same authors, there is some explanation that links the diffracted light with Bragg's equation. Nevertheless, the index of refraction used needs to be clarified. The authors need to clarify that as much as possible.

2) The insensitivity of the diffracted light to the angle is contrary to Bragg's law; please clarify why the colors of the solutions do not change with the angle of incidence or collection of the light.

3) Please clarify the illumination conditions for the dispersions in the insets of  Figures 2-5. 

English is good with only minor issues found (commas, etc)

Reviewer 4 Report

The article titled 'Versatile Double-Bandgap Photonic Crystals of High Color Saturation,' authored by Hao Jiang et al., presents experimental work on dual-bandgap photonic crystals composed of dispersed nanochains. In my assessment, the experimental section is well-detailed and thoroughly discussed. However, the principal conclusions regarding the existence of a 'double band photonic band gap' lack robust support without the inclusion of theoretical work or a physical model. The inquiry into the origin of the photonic bandgap should be addressed more comprehensively. Merely citing the presence of two peaks in reflectance, as opposed to one, as evidence of a double photonic bandgap, is insufficient. Starting from page 6 onward, there is an absence of references that substantiate these conclusions.

My recommendation is "major change".

Minor comments:

page 2 line 53: refracture should be refractive

page 4, line 154. The first paragraph belongs to the template

Round 2

Reviewer 3 Report

I have read the Answers from the authors, and I think the paper is ready for publication.

Author Response

We sincerely appreciate the comments as well as the recognition of our work.

Reviewer 4 Report

Dear Editor and author,

I think that the latest version of the manuscript is more clear and interesting to the potential audience.

I can recommend the paper for publication in Nanomaterials.

Author Response

We are grateful for your feedback and acknowledgment of our efforts.